# Beneficial impact of Ac3IV, an AVP analogue acting specifically at V1a and V1b receptors, on diabetes islet morphology and transdifferentiation of alpha- and beta-cells

Shruti Mohan, Ryan Lafferty, Neil Tanday, Peter R. Flatt, R. Charlotte Moffett, Nigel Irwin◉*

SAAD Centre for Pharmacy and Diabetes, Ulster University, Coleraine, Northern Ireland, United Kingdom

* n.irwin@ulster.ac.uk

**Data Availability Statement:** All relevant data are within the manuscript.

## Abstract

Ac3IV (Ac-CYIQNCPRG-NH$_2$) is an enzymatically stable vasopressin analogue that selectively activates Avpr1a (V1a) and Avpr1b (V1b) receptors. In the current study we have employed streptozotocin (STZ) diabetic transgenic *Ins1*$^{Cre/+}$;*Rosa26-eYFP* and *Glu*$^{CreERT2}$; *Rosa26-eYFP* mice, to evaluate the impact of sustained Ac3IV treatment on pancreatic islet cell morphology and transdifferentiation. Twice-daily administration of Ac3IV (25 nmol/kg bw) to STZ-diabetic *Ins1*$^{Cre/+}$;*Rosa26-eYFP* mice for 12 days increased pancreatic insulin ($p < 0.01$) and significantly reversed the detrimental effects of STZ on pancreatic islet morphology. Such benefits were coupled with increased ($p < 0.01$) beta-cell proliferation and decreased ($p < 0.05$) beta-cell apoptosis. In terms of islet cell lineage tracing, induction of diabetes increased ($p < 0.001$) beta- to alpha-cell differentiation in *Ins1*$^{Cre/+}$;*Rosa26-eYFP* mice, with Ac3IV partially reversing ($p < 0.05$) such transition events. Comparable benefits of Ac3IV on pancreatic islet architecture were observed in STZ-diabetic *Glu*$^{CreERT2}$;*ROSA26-eYFP* transgenic mice. In this model, Ac3IV provoked improvements in islet morphology which were linked to increased ($p < 0.05$–$p < 0.01$) transition of alpha- to beta-cells. Ac3IV also increased ($p < 0.05$–$p < 0.01$) CK-19 co-expression with insulin in pancreatic ductal and islet cells. Blood glucose levels were unchanged by Ac3IV in both models, reflecting the severity of diabetes induced. Taken together these data indicate that activation of islet receptors for V1a and V1b positively modulates alpha- and beta-cell turnover and endocrine cell lineage transition events to preserve beta-cell identity and islet architecture.

## Introduction

Arginine vasopressin (AVP), a peptide secreted from the posterior pituitary, was originally believed to have a primary physiological role in the regulation of body fluid balance and osmolality [1]. However, the presence of functional AVP receptors on pancreatic beta-cells, as well as related positive effects of AVP on beta-cell function and survival [2], confirms an important

**Funding:** Work in the Ulster laboratory was supported by a Ulster University Vice-Chancellors PhD research scholarship (awarded to SM), an early career research award from Diabetes UK (RCM) and Ulster University selective research funding (NI and PRF). The funders had no role in study design, data collection and analysis, decision to publish, or preparation of the manuscript.

**Competing interests:** The authors have declared that no competing interests exist.

endocrine-related function for AVP. In this regard, the biological actions of AVP are associated with activation of three separate G-protein coupled receptors (GPCRs), namely Avpr1a (V1a), Avpr1b (V1b) and Avpr2 (V2) [3]. Whilst V2 receptors are responsible for regulating fluid balance and osmolality [4], V1a and V1b receptors are expressed in metabolically active tissues such as the pancreas [3]. Indeed, a recently characterised enzymatically stable and long-acting AVP analogue, namely Ac3IV, that acts exclusively at V1a and V1b receptors, possesses notable exciting therapeutic potential for diabetes [5].

Accordingly, sustained activation of V1a and V1b receptor pathways by Ac3IV leads to beneficial effects on pancreatic islet architecture, as well as glucose homeostasis and overall metabolic control in high fat fed (HFF) diabetic mice [5]. Interestingly, positive effects of AC3IV on islet architecture were linked to advantageous actions on both islet cell proliferation and apoptosis [5], in direct agreement with previous *in vitro* observations [2]. Although such islet cell turnover effects are undoubtedly important in relation to these structural changes, it is likely that preservation of islet architecture induced by Ac3IV treatment is also related to the processes of islet endocrine cell transdifferentiation [6, 7]. As such, mature islet alpha- and beta-cells have been shown to transition interchangeably between each cell type in response to both induction of diabetes [8] and treatment with established [9–11] or experimental [12–15] anti-diabetic agents.

In the present study, we used transgenic mice with beta-cell lineage tracing capabilities, to investigate the impact of transdifferentiation of beta- to alpha-cells in Ac3IV-induced improvements of pancreatic islet architecture in diabetes. Fully characterised transgenic $Ins1^{Cre/+}$;*Rosa26-eYFP* mice were utilised [10, 11, 13, 14, 16], alongside induction of diabetes and islet damage by multiple low dose streptozotocin (STZ) administration. The subsequent impact of 12 days pharmacological upregulation of V1a and V1b receptor pathways by Ac3IV on beta-cell lineage was then studied. We hypothesised that the pancreatic-related architectural benefits of Ac3IV in diabetes are linked to maintenance of beta-cell identity, alongside the previously observed favourable actions on islet cell proliferation and survival [5]. To examine this theory in more detail, an additional experiment was conducted that involved sustained Ac3IV treatment in STZ-diabetic $Glu^{CreERT2}$;*ROSA26-eYFP* mice, a transgenic mouse model that possesses alpha-cell lineage tracing capabilities [9, 12, 17]. In this way the impact of Ac3IV on alpha- to beta-, as well as beta- to alpha-cell, transdifferentaition could be investigated, which was the primary objective of the current study. Taken together, our datasets suggest that V1a and V1b receptor activation in diabetes induces positive effects on the transdifferentiation of both alpha- and beta-cells, leading to notable benefits on pancreatic islet architecture and beta-cell mass.

## Materials and methods

### Peptides

Ac3IV (Ac-CYIQNCPRG-NH$_2$), a novel enzymatically stable vasopressin analogue with introduction on an N-acetyl group, substitution of F$^3$ for I$^3$ and a disulphide bridge between the two cysteines at position 1 and 6 [5], was obtained from Synpeptide Co. Ltd. (Shanghai, China) at 95% purity. Additional peptide characterisation relating to confirmation of purity and identity was conducted in-house by HPLC and MALDI–ToF MS, as described previously [18].

### Animals

Full details of the generation and characterisation of transgenic $Ins1^{Cre/+}$;*Rosa26-eYFP C57BL/6* and $Glu^{CreERT2}$;*Rosa26-eYFP* C57BL/6 mouse models are provided by Thorens et al., (2015)

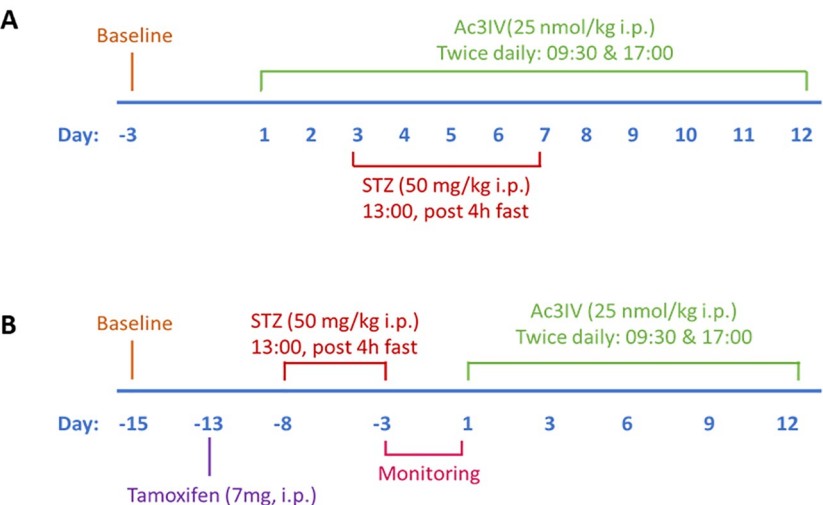

**Fig 1. Experimental timeline for *Ins1Cre/+Rosa26-eYFP* and *GluCreERT2;Rosa26-eYFP* transgenic mouse studies.**
(A) *Ins1Cre/+;Rosa26-eYFP* mice (n = 6) received twice daily treatment with saline vehicle (0.9% NaCl) or Ac3IV (25 nmol/kg bw, i.p.) 3 days prior to induction of insulin-deficient diabetes by STZ. (B) The ability of twice daily treatment with Ac3IV (25 nmol/kg bw, i.p.) to reverse beta-cell loss in STZ-induced diabetes was examined in *GluCreERT2*; *Rosa26-eYFP* mice (n = 6).

and Campbell et al., (2020), respectively [16, 17]. All mice were bred in-house with PCR genotyping for each colony employed as previously described by our laboratory [10, 12]. Experiments were carried out under the UK Animals (Scientific Procedures) Act 1986 and EU Directive 2010/63EU as well as being approved by the local Ulster University Animal Welfare and Ethical Review Body (AWERB). Animals were used at 12–14 weeks of age and were maintained in an environmentally controlled unit at 22 ± 2 ˚C with a 12 h dark and light cycle and given *ad libitum* access to standard rodent diet (10% fat, 30% protein and 60% carbohydrate; Trouw Nutrition, Northwich, UK) and drinking water.

## Experimental protocols

*Ins1Cre/+;Rosa26-eYFP* mice (n = 6) received twice-daily (09:00 and 17:00 h) treatment with either saline vehicle (0.9% (w/v) NaCl) or Ac3IV (25 nmol/kg bw) 3 days prior to induction of insulin-deficient diabetes, and throughout the full duration of the study (Fig 1A). The peptide dosing regimen was based on previous studies with Ac3IV and related peptides [5, 19]. Diabetes was induced by low dose STZ injection (50 mg/kg body weight, i.p.) for 5 consecutive days. In a separate series of experiments, the ability of Ac3IV to reverse beta-cell loss in STZ-induced diabetes was examined in *GluCreERT2;Rosa26-eYFP* mice. These mice were administered tamoxifen (7 mg/mouse bw, i.p.) to induce expression of the alpha-cell fluorescent lineage marker protein, 5 days prior to induction of insulin-deficient diabetes by multiple low dose STZ injection, as described above. Upon confirmation of hyperglycaemia and diabetes development, *GluCreERT2;Rosa26-eYFP* mice received twice-daily (09:00 and 17:00 h) treatment with either saline vehicle (0.9% (w/v) NaCl) or Ac3IV (25 nmol/kg bw) for 12 days (Fig 1B). For all experiments, body weight, cumulative food intake and blood glucose were assessed at regular intervals. At the end of the treatment period, non-fasting plasma insulin and glucagon concentrations were determined. At termination, animals were killed by cervical dislocation and pancreatic tissues excised, divided longitudinally, and processed for either determination of

pancreatic hormone content following acid/ethanol protein extraction or fixed in 4% PFA for 48 h at 4 ˚C for histological analysis [20].

## Immunohistochemistry

Fixed tissues were processed and embedded in paraffin wax blocks using an automated tissue processor (Leica TP1020, Leica Microsystems) and 5 μm sections cut on a microtome (Shandon Finesse 325, Thermo Scientific). Slides were dewaxed by immersion in xylene and rehydrated through a series of ethanol solutions of reducing concentration (100–50%). Heat-mediated antigen retrieval was then carried out in citrate buffer. Sections were blocked in 4% BSA solution before 4˚C overnight incubation with appropriate primary antibodies including insulin (1:400; Abcam, ab6995), glucagon (1:400; raised in-house, PCA2/4), GFP (1:1000; Abcam, ab5450), Ki-67 (1:500; Abcam, ab15580) or CK-19 (1:500; Abcam, ab15463). The glucagon primary antibody (PCA2/4) was raised in house in guinea-pigs immunised with porcine glucagon-carbodiimide-albumin conjugates [21], and specificity confirmed in our previous studies [22]. In this respect, CK-19 is normally expressed in pancreatic exocrine tissue with such cells shown to be capable of developing into endocrine insulin-positive islet cells [23]. Slides were then rinsed in PBS and incubated for 45 min at 37˚C with appropriate Alexa Fluor secondary antibodies (1:400; Invitrogen, Alexa Fluor 488 for green or 594 for red, Invitrogen). The following secondary antibodies were employed, as appropriate, goat anti-mouse Alexa Fluor 488, goat anti-mouse Alexa Fluor 594, goat anti-guinea pig Alexa Fluor 488, goat anti-guinea pig Alexa Fluor 594, donkey anti-goat 488 and goat anti-rabbit Alexa Fluor 488. Slides were finally incubated with DAPI for 15 min at 37˚C, and then mounted for imaging using a fluorescent microscope (Olympus model BX51) fitted with DAPI (350 nm) FITC (488 nm) and TRITC (594 nm) filters and a DP70 camera adapter system [10].

## Image analysis

Islet parameters, including islet, beta- and alpha-cell areas, were analysed using the Cell$^F$ imaging software and the closed loop polygon tool (Olympus Soft Imaging Solutions). For transdifferentiation cells co-expressing both insulin and GFP (insulin$^{+ve}$, GFP$^{+ve}$ cells), cells expressing insulin with no GFP (insulin$^{+ve}$, GFP$^{-ve}$ cells), cells expressing GFP without insulin (insulin$^{-ve}$, GFP$^{+ve}$ cells), cells expressing glucagon without GFP (glucagon$^{+ve}$, GFP$^{-ve}$ cells) along with cells co-expressing GFP and glucagon (glucagon$^{+ve}$, GFP$^{+ve}$ cells) were analysed, as appropriate. In addition, in *Ins1$^{Cre/+}$*;*Rosa26-eYFP* mice, islet cell apoptosis was determined using co-expression of TUNEL with either insulin or glucagon. Similarly, islet cell proliferation was also assessed using Ki-67 staining and co-expression with either insulin or glucagon. All cell counts were determined in a blinded manner with >50 islets analysed per treatment group.

## Biochemical analyses

Blood samples were collected from the cut tail vein of animals. Blood glucose was measured using a portable Ascencia Contour blood glucose meter (Bayer Healthcare, Newbury, Berkshire, UK). For plasma insulin and glucagon, blood was collected in chilled fluoride/heparin coated microcentrifuge tubes (Sarstedt, Numbrecht, Germany) and centrifuged using a Beckman micro-centrifuge (Beckman Instruments, Galway, Ireland) for 10 min at 12,000 rpm. Plasma was removed and stored at −20 ˚C, until required for analysis. For hormone content, snap frozen pancreatic tissues were homogenised in acid/ethanol (75% (v/v) ethanol, distilled water and 1.5% (v/v) 12 M HCl) and protein extracted in a pH neutral TRIS buffer, with protein content determined using Bradford reagent (Sigma-Aldrich). Plasma and pancreatic insulin content were determined by an in-house insulin RIA [24], whilst plasma and pancreatic

glucagon content were assessed by a commercially available ELISA kit (glucagon chemilumi-nescent assay, EZGLU-30K, Millipore) following the manufacturer's guidelines.

## Statistics

Data were analysed using GraphPad PRISM 5.0, with data presented as mean ± SEM. Comparative analyses between groups of mice were carried out using a one-way ANOVA with a Bonferroni p*ost hoc* test or a two-way repeated measures ANOVA with a Bonferroni p*ost hoc* test, as appropriate. Results were deemed significant if p<0.05.

## Results

### Effects of Ac3IV on metabolic indices and pancreatic hormone content in STZ-diabetic *Ins1*^CRE/+;Rosa26-eYFP mice

STZ induced a significant (p<0.01) reduction in body weight change (Fig 2A), which was linked to decreased (p<0.05 –p<0.001) energy intake (Fig 2B). Ac3IV treatment had no significant impact on body weight change or energy intake (Fig 2A and 2B). As expected, STZ also increased (p<0.001) blood glucose levels during the 12 day study, which was not affected by Ac3IV (Fig 2C). Neither STZ nor Ac3IV intervention altered non-fasting insulin or glucagon concentrations on day 12 (Fig 2D and 2E). In terms of pancreatic insulin content, as expected STZ decreased (p<0.001) this parameter whereas Ac3IV treatment was associated with a significant (p<0.01) increase in pancreatic insulin content compared to STZ controls (Fig 2F). Pancreatic glucagon was unaltered in STZ mice compared to lean control mice, but Ac3IV evoked a significant (p<0.05) decrease when compared to STZ-diabetic controls (Fig 2G).

### Effects of Ac3IV on pancreatic islet morphology in STZ- diabetic *Ins1*^CRE/+; Rosa26-eYFP mice

Representative images for pancreatic islets stained for insulin and glucagon are shown in Fig 3A. STZ significantly decreased (p<0.001) pancreatic islet area compared to saline controls

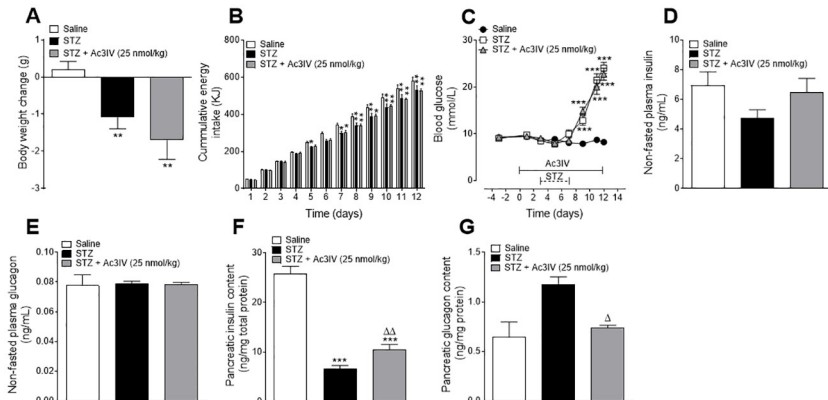

**Fig 2. Effects of Ac3IV treatment on body weight, blood glucose, energy intake as well as circulating and pancreatic insulin and glucagon in STZ-diabetic Ins1Cre/+Rosa26-eYFP mice.** Body weight change (A), cumulative energy intake (B) and circulating blood glucose (C) were measured during twice daily treatment with saline vehicle (0.9% NaCl) or Ac3IV (25 nmol/kg bw, i.p.) for 12 days in STZ- diabetic Ins1Cre/+Rosa26-eYFP transgenic mice. (D-G) Plasma and pancreatic insulin (D,F) and glucagon (E,G) levels were assessed on day 12. Solid lines parallel to x-axis indicate peptide treatment while dashed lines represent days of diabetes induction by STZ (C). Values are mean ± SEM for n = 6 mice. Values are mean ± SEM for n = 6 mice. *p<0.05, **p<0.01, ***p<0.001 compared to saline vehicle. Δp<0.05, ΔΔp<0.01 compared to STZ control.

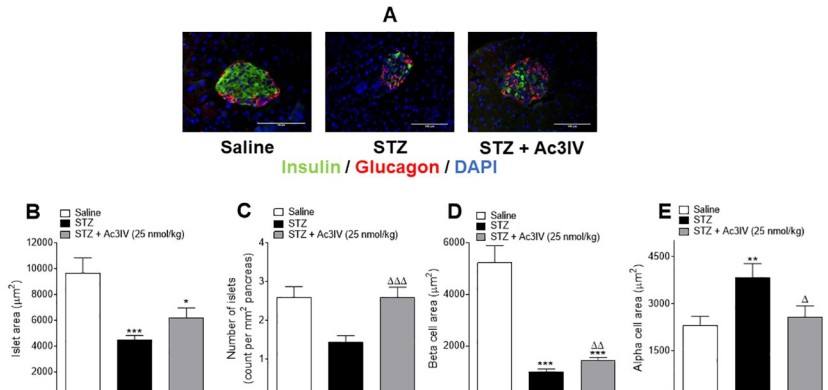

**Fig 3. Effects of Ac3IV treatment on pancreatic morphology in STZ-diabetic *Ins1Cre^{/+}Rosa26-eYFP* mice.**
Parameters were assessed following 12 days twice-daily treatment with saline (0.9% NaCl) vehicle or Ac3IV (25 nmol/kg bw, i.p.) in STZ-diabetic *Ins1Cre^{/+}Rosa26-eYFP* transgenic mice. Representative images (40X) of stained islets are provided in panel (A). Islet area (B) number of islets/mm² of pancreas (C), beta- (D) and alpha-cell areas (E) were assessed using Cell^F imaging software and the closed loop polygon tool. Values are mean ± SEM for n = 6 mice. *p<0.05, **p<0.01, ***p<0.001 compared to saline vehicle. ^Δp<0.05, ^ΔΔp<0.01, ^ΔΔΔp<0.001 compared to STZ-diabetic controls.

(Fig 3B), and Ac3IV had a tendency to increase this parameter (Fig 3B). Pancreatic islet numbers were reduced (p<0.05) by STZ, with Ac3IV fully reversing this effect (p<0.001) and returning islet numbers to lean control levels (Fig 3C). Similarly, STZ decreased (p<0.001) beta-cell area with Ac3IV significantly (p<0.01) countering this effect (Fig 3D). STZ also increased (p<0.05) alpha-cell area and Ac3IV fully reversed this effect (Fig 3E).

## Effects of Ac3IV on beta- to alpha-cell transdifferentiation in STZ-diabetic *Ins1*^{CRE/+};Rosa26-eYFP mice

Representative images for pancreatic islets co-stained for GFP with insulin or glucagon are shown in Fig 4A and 4B. STZ did not induce any changes in the percentage of insulin^{+ve}, GFP^{-ve} islet cells (Fig 4C). However, treatment with Ac3IV significantly (p<0.01) increased insulin^{+ve}, GFP^{-ve} cells (Fig 4C). STZ mice presented with an increased percentage of insulin^{-ve}, GFP^{+ve} cells, which was significantly (p<0.05) reduced by Ac3IV (Fig 4D). In harmony with this, the increase of glucagon^{+ve}, GFP^{+ve} cells in STZ mice was reduced (p<0.05) by Ac3IV treatment (Fig 4E).

## Effects of Ac3IV on alpha and beta-cell proliferation and apoptosis, as well as ductal cell transdifferentiation, in STZ-diabetic *Ins1*^{CRE/+};Rosa26-eYFP mice

Beta-cell proliferation was not significantly altered in STZ mice (Fig 5A), but Ac3IV significantly (p<0.01) increased this parameter (Fig 5A). Whilst alpha-cell proliferation was increased in STZ-diabetic mice, this elevation was not significant when compared to lean controls (Fig 5B). However, Ac3IV intervention decreased (p<0.05) alpha-cell proliferation when compared to STZ-diabetic control mice (Fig 5B). In terms of beta-cell apoptosis, STZ mice had dramatically increased (p<0.001) apoptotic rates which were reduced (p<0.05) by Ac3IV (Fig 5C). Alpha-cell apoptotic rates were not significantly different from respective lean controls in

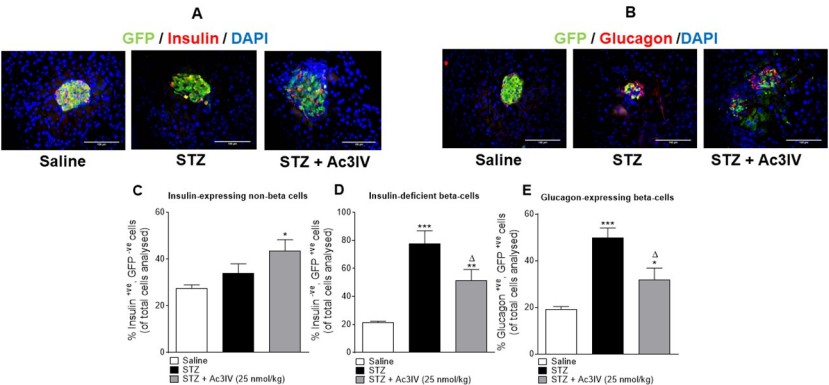

**Fig 4. Effects of Ac3IV treatment on pancreatic islet cell lineage in STZ-diabetic *Ins1Cre/+Rosa26-eYFP* mice.**
Parameters were assessed following 12 days twice-daily treatment with saline (0.9% NaCl) vehicle or Ac3IV (25 nmol/kg bw, i.p.) in STZ-diabetic *Ins1Cre/+Rosa26-eYFP* transgenic mice. Representative images (40X) of pancreatic islets depicting co-localisation of GFP (green) with either insulin or glucagon (red) are shown in panels (A&B). Numbers of insulin+ve, GFP-ve (C), insulin-ve, GFP+ve (D) and glucagon+ve, GFP+ve (E) islet stained cells were assessed utilising the cell counting function within Cell^F imaging software. Values are mean ± SEM for n = 6 mice. *p<0.05, **p<0.01, ***p<0.001 compared to saline vehicle. ^Δp<0.05 compared to STZ-diabetic controls.

STZ mice (Fig 5D), although Ac3IV increased (p<0.05) alpha-cell apoptosis when compared to STZ-diabetic controls (Fig 5D). With regards to pancreatic ductal cells, STZ significantly (p<0.05) reduced the percentage of insulin positive ductal cells, an effect that was fully reversed by Ac3IV treatment (Fig 5E). In islets, the co-expression of insulin and CK-19 was reduced (p<0.001) by STZ, but restored to levels similar to lean control mice by Ac3IV (Fig 5F).

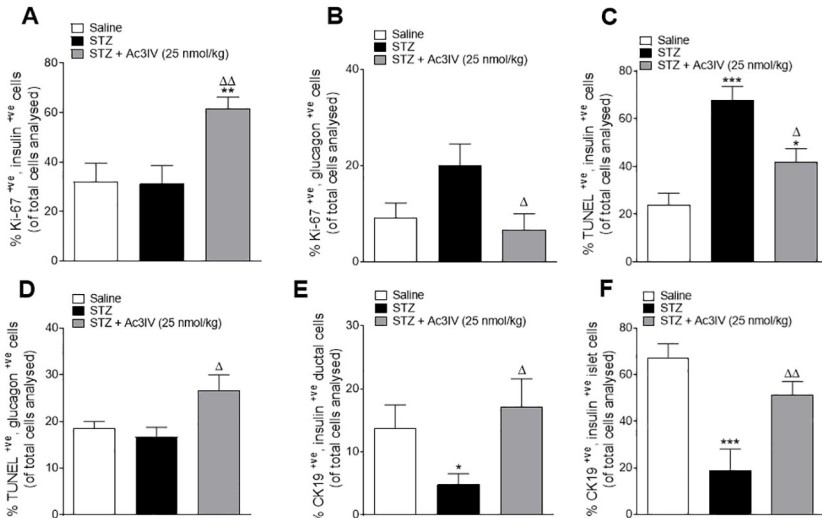

**Fig 5. Effects of Ac3IV treatment on islet cell turnover and expression of ductal cell markers in STZ-diabetic *Ins1Cre/+Rosa26-eYFP* mice.** Parameters were assessed following 12 days twice-daily treatment with saline (0.9% NaCl) vehicle or Ac3IV (25 nmol/kg bw, i.p.) in STZ- diabetic *Ins1Cre/+Rosa26-eYFP* transgenic mice. Beta- and alpha-cell proliferation (A,B) and apoptosis (C,D) were assessed by Ki67 or TUNEL co-staining with insulin/glucagon, respectively. CK-19 co-localisation with insulin in pancreatic ductal (E) and islet (F) cells. Values are mean ± SEM for n = 6 mice. *p<0.05, **p<0.01, ***p<0.001 compared to saline vehicle. ^Δp<0.05 and ^ΔΔp<0.01 compared to STZ-diabetic controls.

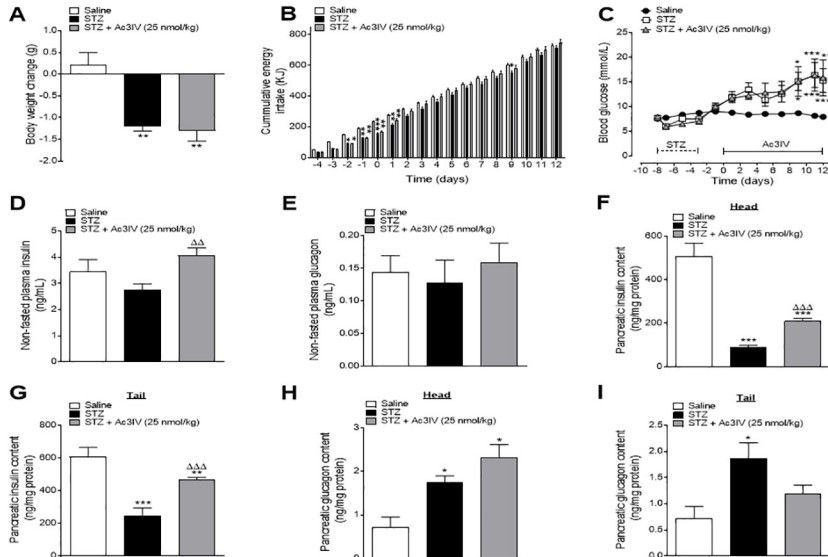

**Fig 6. Effects of Ac3IV treatment on body weight, blood glucose, plasma insulin and pancreatic hormone content in STZ-diabetic GluCreERT2;Rosa26-eYFP mice.** Body weight change (A), cumulative energy intake (B) and circulating blood glucose (C) were measured during twice daily treatment with saline (0.9% NaCl) vehicle or Ac3IV (25 nmol/kg bw, i.p.) for 12 days in STZ-diabetic GluCreERT2;Rosa26-eYFP transgenic mice. Plasma insulin (D) and glucagon (E), as well as pancreatic hormone content in the head (F,H) and tail (G,I) of the pancreas were measured on day 12. Solid lines parallel to x-axis indicate peptide treatment while dashed lines represent days of diabetes induction by STZ (C). Values are mean ± SEM for n = 6 mice. *p<0.05, **p<0.01, ***p<0.001 compared to saline vehicle. ΔΔp<0.01 and ΔΔΔp<0.001 compared to STZ-diabetic controls.

## Effects of Ac3IV on metabolic indices and pancreatic hormone content in STZ-diabetic $Glu^{CreERT2}$;$Rosa26$-eYFP mice

The remarkable increase of alpha-cells in STZ mice together with positive effects of Ac3IV on beta-cell parameters begged the question of whether Ac3IV could stimulate alpha- to beta-cell transdifferentiation. In complete harmony with observations in $Ins1^{CRE/+}$;Rosa26-eYFP mice, STZ induced a significant (p<0.01) reduction in body weight change in $Glu^{CreERT2}$; $Rosa26$-eYFP mice (Fig 6A), linked to decreased (p<0.05 –p<0.001) energy intake (Fig 6B). STZ also increased (p<0.05 –p<0.001) individual daily blood glucose levels which was unaltered by Ac3IV treatment (Fig 6C). Interestingly, at the end of the study, plasma insulin levels were not altered by STZ, but Ac3IV increased (p<0.01) this parameter when compared to STZ-diabetic controls (Fig 6D). Neither STZ nor Ac3IV altered circulating glucagon concentrations (Fig 6E). However, STZ reduced pancreatic insulin content (p<0.001) in both the head and tail of the pancreas, which were both significantly (p<0.001) elevated by Ac3IV (Fig 6F and 6G). Pancreatic glucagon content was elevated (p<0.05) in the head and tail portions of the pancreas (Fig 6H and 6I), with Ac3IV fully reversing this effect in the pancreatic tail (Fig 6I).

## Effects of Ac3IV on pancreatic islet morphology in STZ-diabetic $Glu^{CreERT2}$;$Rosa26$-eYFP mice

Ac3IV countered the effects of STZ on islet morphology and significantly increased islet- and beta-cell areas in the head and tail of the pancreas (p<0.05–0.001; Fig 7A–7D). Interestingly, alpha-cell area was also increased in both portions of the pancreas in Ac3IV treated mice when

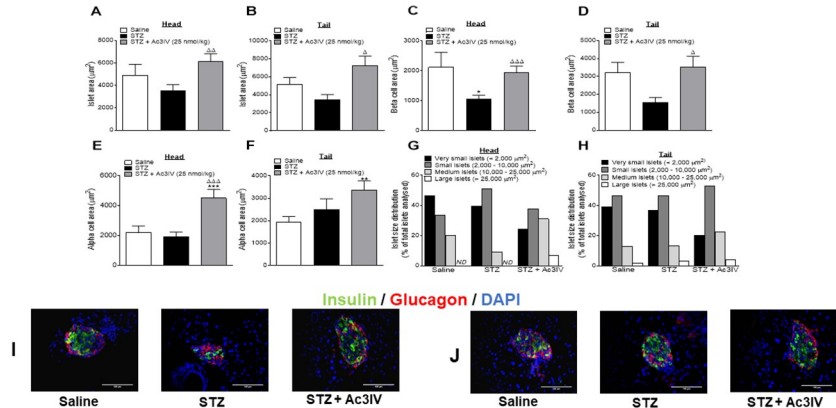

**Fig 7. Effects of Ac3IV treatment on pancreatic morphology in STZ-diabetic GluCreERT2;Rosa26-eYFP mice.**
Parameters were assessed in the head (A,C,E,G) and tail (B,D,F,H) of the pancreas following 12 days twice-daily treatment with saline (0.9% NaCl) vehicle or Ac3IV (25 nmol/kg bw, i.p.) in STZ-diabetic GluCreERT2;Rosa26-eYFP transgenic mice. Islet (A,B), beta- (C,D) and alpha-cell (E,F) areas were assessed using CellF imaging software and the closed loop polygon tool. Islet size distributions in the head (G) and tail (H) of the pancreas are also shown. Representative images (40X) of stained islets are provided in panels I and J. Values are mean ± SEM for n = 6 mice. *p<0.05, **p<0.01, ***p<0.001 compared to saline vehicle. Δp<0.05, ΔΔp<0.01 and ΔΔΔp<0.001 compared to STZ-diabetic controls. ND—not detected.

compared to lean controls (p<0.001 and p<0.01, respectively), and in the head of the pancreas (p<0.001) when compared to STZ-diabetic mice (Fig 7E and 7F). The number of very small (<2,000 μm$^2$), small (2,000–10,000 μm$^2$) and medium (10,000–25,000 μm$^2$) sized islets was similar across all groups of mice in the head of the pancreas, but larger (>25,000 μm$^2$) sized islets were only detectable in Ac3IV treated mice (Fig 7G). Islet size distribution was not noticeably different between all the various groups in pancreatic tail, but Ac3IV treated mice did appear to have reduced numbers of very small (<2,000 μm$^2$) islets, with larger sized islets being more prevalent (Fig 7H). Representative images immunohistochemically stained for insulin and glucagon from the head and tail of the pancreas for each treatment group are shown in Fig 7I and 7J, respectively.

### Effects of Ac3IV on alpha to beta-cell transdifferentiation in STZ-diabetic $Glu^{CreERT2}$;Rosa26-eYFP mice

STZ did not change the percentage of insulin$^{+ve}$, GFP$^{+ve}$ or glucagon$^{+ve}$, GFP$^{-ve}$ islet cells in the head and tail of the pancreas in $Glu^{CreERT2}$;Rosa26-eYFP mice (Fig 8A–8D). However, treatment with Ac3IV significantly increased (p<0.05 –p<0.01) insulin$^{+ve}$, GFP$^{+ve}$ cells in both the head and tail when compared to saline control and STZ-diabetic mice (Fig 8A and 8B). Although there was no change in the percentage of glucagon$^{+ve}$, GFP$^{-ve}$ cells in the head of the pancreas between the groups of mice (Fig 8C), Ac3IV significantly (p<0.05) decreased this cell population in the tail of the pancreas when compared to STZ-diabetic control mice (Fig 8D). Representative images for pancreatic islets co-stained for GFP with insulin or glucagon are shown in Fig 8E and 8F.

### Discussion

As expected, induction of diabetes with multiple low dose STZ in $Ins1^{CRE/+}$;Rosa26-eYFP mice resulted in detrimental changes in pancreatic islet morphology and characteristic metabolic derangements [25]. These mice exhibited marked decreases in islet number and size as well as

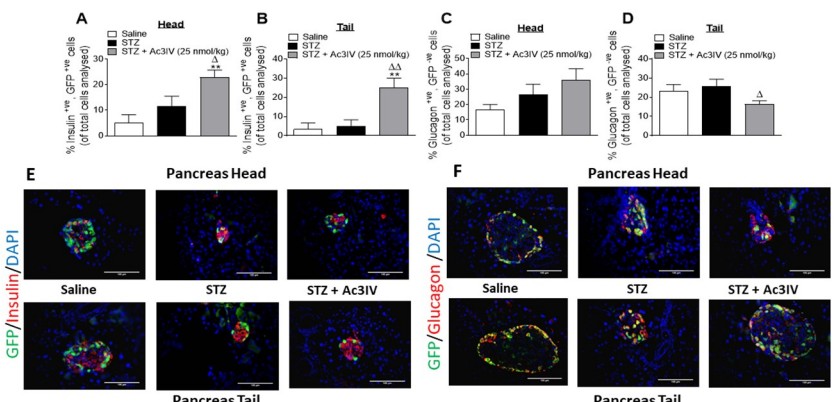

**Fig 8. Effects of Ac3IV treatment on pancreatic islet cell lineage in STZ—diabetic $Glu^{CreERT2}$;$Rosa26$-$eYFP$ mice.**
Parameters were assessed following 12 days twice-daily treatment with saline (0.9% NaCl) vehicle or Ac3IV (25 nmol/kg bw, i.p.) in STZ-diabetic $Glu^{CreERT2}$;$Rosa26$-$eYFP$ transgenic mice. Numbers of insulin+ve, GFP+ve (A,B) and glucagon+ve, GFP-ve (C,D) islet stained cells in the head (A,C) and tail (B,D) of the pancreas are shown as percentages of total cell counts. Representative images (40X) of islets from both the head and tail of the pancreas depicting co-localisation of GFP (green) with either insulin or glucagon (red) are shown in panels E&F. Values are mean ± SEM for n = 6 mice. **p<0.01 compared to saline vehicle. $^{\Delta}$p<0.05 and $^{\Delta\Delta}$p<0.01 compared to STZ-diabetic controls.

beta-cell area, together with expansion of alpha-cells which culminated in overt hyperglycae-mia. Changes in islet architecture confirm appropriateness of the $Ins1^{CRE/+}$;$Rosa26$-$eYFP$ transgenic mouse model to investigate islet cell lineage adaptations in response to sustained V1a and V1b receptor activation by Ac3IV in degenerative diabetes.

In keeping with earlier observations in HFF mice [5], Ac3IV treatment evoked a marked reversal of the negative impact of STZ on pancreatic islet architecture. Thus, Ac3IV-induced significant increases of beta-cell area, associated with positive effects on beta-cell proliferation and protection against apoptosis. Interestingly, although these mice had relative increases in alpha-cell area, that were fully corrected by Ac3IV treatment, circulating glucagon was unaffected. This contrasts with the view that V1b receptor activation elicits glucagonotropic actions [25], but the mismatch most likely reflects the regulated secretion of islet hormones [26]. Indeed, elevation of pancreatic insulin by Ac3IV in $Ins1^{CRE/+}$;$Rosa26$-$eYFP$ mice also lacked obvious correlation with circulating insulin. More notably, the decreases of alpha-cell area and pancreatic glucagon were accompanied by enhanced alpha-cell apoptosis and tendency for decreased alpha-cell proliferation. This is interesting given that V1a and V1b receptor activation is generally associated with pro-survival effects [2], linked to Gq receptor coupling and activation of the phospholipase C (PLC) pathway. Effects of Ac3IV in the current setting may simply represent the capacity for Ac3IV to reverse the detrimental effects of STZ on pancreatic islet architecture. Indeed, similar effects of Ac3IV to decrease alpha-cell growth have been demonstrated in HFF mice [5].

Given the well characterised roles for V1a and V1b receptors in pancreatic endocrine function [19], sustained activation of these receptors might help counter disturbed pancreatic islet integrity via positive effects on islet cell transdifferentiation [6, 7]. Indeed, islet cell lineage tracing in $Ins1^{CRE/+}$;$Rosa26$-$eYFP$ transgenic mice revealed that even healthy control mice exhibited some degree of beta-cell identity loss together with transitioning to glucagon positive cells. Transdifferentiation of beta- to alpha-cells therefore appears to be a natural phenomenon that is amplified by diabetes [27]. Consistent with this view, in untreated STZ-diabetic mice there were clear parallels between alterations in the numbers of insulin-ve, GFP+ve and glucagon+ve,

GFP$^{+ve}$ islet cells. Importantly, Ac3IV was able to fully, or partially, oppose such detrimental islet cell transdifferentiation events. This included decreased numbers of insulin$^{-ve}$, GFP$^{+ve}$ beta-cells losing insulin expression, increased numbers of non-beta insulin$^{+ve}$, GFP$^{-ve}$ cells expressing insulin plus decreased transdifferentiation of beta-cells to glucagon expressing glucagon$^{+ve}$, GFP$^{+ve}$ cells. A related study, also employing $Ins1^{CRE/+}$;$Rosa26-eYFP$ transgenic mice, recently examined the positive impact of clinically approved incretin enhancers, liraglutide and sitagliptin, on islet cell lineage in diabetes [10]. Although somewhat challenging to directly compare the relative magnitude of incretin signalling induced benefits with the current data, it does appear that Ac3IV has equivalent, if not superior, efficacy than liraglutide or sitagliptin in terms of limiting beta-cell transdifferentaition in diabetes [10]. Since V1b receptor activation has also been reported to induce secretion of GLP-1 [28], this might also be a factor in mediating the overall effect of Ac3IV. It is also notable that the islet benefits of Ac3IV were independent of changes in glycaemic status which were minimal in the STZ mice. Assessment of glucose and insulin tolerance would also have been useful to determine the positive impact of Ac3IV on overall metabolism, but our primary objective in the transgenic mouse models utilised for the current study was evaluation of effects on islet cell lineage. Moreover, benefits of Ac3IV on glucose homeostasis and insulin action have previously been confirmed in HFF obese mice [5].

To confirm positive effects of Ac3IV on islet cell transdifferentaition, complementary studies were performed in STZ-induced diabetic $Glu^{CreERT2}$;$Rosa26-eYFP$ transgenic mice, with alpha-cell linage tracking capabilities [17]. The protocol used differed slightly in these experiments in that Ac3IV was administered after induction of diabetes, when changes in islet cell populations were already established. This approach enabled terminal observations on islets with relative enrichment of alpha cells due to severe beta-cell loss. In agreement with observations in $Ins1^{CRE/+}$;$Rosa26-eYFP$ mice, repeated low-dose STZ induced abnormalities that were significantly improved by Ac3IV intervention. Given that pancreatic islets are more concentrated and richer in alpha-cells in the tail of the pancreas in rodents [29, 30], we analysed islet morphology separately in the head and tail regions. Furthermore, in terms of effects of Ac3IV on islet cell lineage it may be of interest to note that the head and the tail of the pancreas have different developmental origins. As such, the head of the pancreas is formed from the dorsal and ventral pancreatic bud, whereas the tail is derived from the ventral pancreatic bud [30]. Moreover, there is a suggestion that islets from the head of the pancreas have a greater insulin secretory capacity than those originating solely from the ventral bud region [31]. Beneficial effects of Ac3IV on pancreatic morphology were relatively consistent across both portions of the pancreas, preserving beta-cell mass and increasing pancreatic and circulating insulin. As observed with $Ins1^{CRE/+}$;$Rosa26-eYFP$ mice, beneficial reduction of alpha-cells, particularly in the alpha-cell rich pancreatic tail, did not translate to parallel changes in pancreatic and plasma glucagon. However, there was clear evidence for increased alpha- to beta-cell transdifferentaition in both regions [7], correlating well with the observed increase of insulin$^{+ve}$, GFP$^{-ve}$ cells in Ac3IV treated $Ins1^{CRE/+}$;$Rosa26-eYFP$ mice. In addition, Ac3IV decreased the emergence of newly formed glucagon expressing alpha-like glucagon$^{+ve}$, GFP$^{-ve}$ cells in the tail of the pancreas.

Although these additional islet cell lineage data in $Glu^{CreERT2}$;$Rosa26-eYFP$ mice are extremely useful and corroborate observations on beneficial Ac3IV-induced changes in $Ins1^{CRE/+}$;$Rosa26-eYFP$ transgenic mice, it is worth noting that $Glu^{CreERT2}$ and $Ins1^{Cre/+}$ are not analogous lineage tracing models. As such, the fluorescent transgene label in $Ins1^{CRE/+}$;$Rosa26-eYFP$ mice will immediately be present upon production of $Ins1$ in the beta-cell, whereas in $Glu^{CreERT2}$;$Rosa26-eYFP$ mice fluorescent labelling is only apparent when alpha-cells are concomitantly induced by tamoxifen [16, 17]. In addition, commencement of the

Ac3IV therapeutic regimen in relation to onset of STZ-induced diabetes and was not identical in *Ins1*CRE/+;*Rosa26-eYFP* and *Glu*CreERT2;*Rosa26-eYFP* mice, but the consistency of our observations in the two transgenic models indicates sound validity of our observations. However, the established detrimental effects of STZ on islet architecture were apparent to a greater extent in *Ins1*CRE/+;*Rosa26-eYFP* mice, in agreement with recent observations of a more severe STZ-induced phenotype in these mice compared with *Glu*CreERT2;*Rosa26-eYFP* diabetic mice [15]. This difference was suggested to be linked to variations in STZ susceptibility between the two strains of mice. Furthermore, although beyond the scope of the current study, consideration of levels of specific alpha- and beta-cell transcription factors such as aristaless-related homeobox (Arx), pancreatic and duodenal homeobox 1 (Pdx-1) or NK6 homeobox 1 (NKX6.1) would be useful to further assess islet cell lineage fate [7].

Further to positive impacts on endocrine islet cell transdifferentaition and beta-cell de-differentiation, an additional effect of Ac3IV to preserve normal islet structure in diabetes could be linked to the morphogenesis of pluripotent pancreatic exocrine ductal cells towards insulin positive islet cells [23]. As such, lineage studies would suggest that pancreatic ductal epithelium cells can represent progenitors for islet endocrine cells [32, 33], although others have contested this pathway [34, 35]. Despite this, there was clear evidence in our study for Ac3IV to increase the percentage of ductal and islet cells co-expressing CK-19 and insulin in STZ-diabetic *Ins1*CRE/+;*Rosa26-eYFP* mice, suggesting that these cells are involved in replenishing beta-cells from a non-endocrine source [36]. In agreement, Ac3IV also increased the number of pancreatic islets per mm$^2$ in STZ-diabetic *Ins1*CRE/+;*Rosa26-eYFP* mice.

In conclusion, the present study demonstrates that sustained V1a and V1b receptor activation by Ac3IV preserves pancreatic islet structure in diabetes by positively influencing the transition of both islet alpha- and beta-cells [16, 17], leading to enhanced metabolic control. As observed previously [5], Ac3IV also exerted beneficial actions on islet cell proliferation and protection against apoptosis to additionally conserve normal islet architecture. Taken together, our new data demonstrate, for the first time, that pancreatic islet benefits of V1a and V1b receptor activation in diabetes are linked to improvements in islet morphology and both alpha- and beta-cell transdifferentiation.

## Acknowledgments

The authors thank Professor F Reimann (University of Cambridge) for the donation of breeding pairs of *Glu*CreERT2;*Rosa26-eYFP* mice.

## Author Contributions

**Conceptualization:** Peter R. Flatt, Nigel Irwin.

**Data curation:** Shruti Mohan, Ryan Lafferty, Neil Tanday.

**Formal analysis:** Shruti Mohan, Neil Tanday, Nigel Irwin.

**Funding acquisition:** Peter R. Flatt, R. Charlotte Moffett, Nigel Irwin.

**Investigation:** Shruti Mohan, Ryan Lafferty, Neil Tanday.

**Methodology:** Ryan Lafferty, Neil Tanday.

**Project administration:** Peter R. Flatt, R. Charlotte Moffett, Nigel Irwin.

**Resources:** Peter R. Flatt.

**Software:** Neil Tanday.

**Supervision:** Peter R. Flatt, R. Charlotte Moffett, Nigel Irwin.

**Validation:** Shruti Mohan, Ryan Lafferty, Neil Tanday, Peter R. Flatt, Nigel Irwin.

**Visualization:** Neil Tanday, R. Charlotte Moffett.

**Writing – original draft:** Shruti Mohan, Ryan Lafferty, Neil Tanday, Peter R. Flatt, Nigel Irwin.

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
