## [Decision Letter · Decision Letter 0]

18 Aug 2021

PONE-D-21-20871

Beneficial impact of Ac3IV, an AVP analogue acting specifically at V1a and V1b receptors, on diabetes islet morphology and transdifferentiation of alpha- and beta-cells

PLOS ONE

Dear Dr. Irwin,

Thank you for submitting your manuscript to PLOS ONE. After careful consideration, we feel that it has merit but does not fully meet PLOS ONE’s publication criteria as it currently stands. Therefore, we invite you to submit a revised version of the manuscript that addresses the points raised during the review process.

Your manuscript has been carefully evaluated by two external reviewers with expertise in this field. They raised several concerns below. Especially, there is discrepancy in quantification data of immunohistochemistry. Expression of transcription factors critical for transdifferentiation is not shown. These critical suggestions should be addressed for further consideration. Because conclusions are not presented in an appropriate fashion and are not supported by the data, this manuscript cannot be recommended for publication in PLoS ONE in its current form. 

We look forward to receiving your revised manuscript.

Kind regards,

Wataru Nishimura, M.D., Ph.D.

Academic Editor

PLOS ONE

Reviewers' comments:

Reviewer's Responses to Questions

**Comments to the Author**

1. Is the manuscript technically sound, and do the data support the conclusions?

Reviewer #1: Yes

Reviewer #2: Yes

2. Has the statistical analysis been performed appropriately and rigorously? 

Reviewer #1: Yes

Reviewer #2: Yes

3. Have the authors made all data underlying the findings in their manuscript fully available?

Reviewer #1: Yes

Reviewer #2: Yes

4. Is the manuscript presented in an intelligible fashion and written in standard English?

Reviewer #1: Yes

Reviewer #2: Yes

5. Review Comments to the Author

Reviewer #1: Shruit et al reported that Ac3IV, an AVP analog that acts specifically on V1a and V1b receptors, induces positive effects on the transdifferentiation of both alpha and beta cells, leading to benefits on islet structure and beta cell mass.

It is new important knowledge that V1a and V1b receptor activation are related to improvement of both transdifferentiation of both alpha and beta cell.

There are, however, minor issues that should be addressed.

major points:

1. The authors measured blood glucose and insulin levels as the effect of Ac3 IV on blood glucose level, glucose tolerance test and insulin tolerance test should be performed to evaluate glucose metabolism in more detail.

2. The authors used insulin and glucagon-positive cells to evaluate transdifferentiation between alpha and beta cells, but in order to obtain more detailed results, other markers such as PDX1, Nkx6.1, and Arx Expression should be assessed by staining or gene expression.

Reviewer #2: In this study, the effects of a vasopressin analogue Ac3IV administration on islet cell morphology, turnover and lineage transition are investigated in detail based on a previous report by the authors that Ac3IV has an anti-diabetic effect in mice. The authors suggest that Ac3IV administration improves the islet morphological changes induced by streptozotocin by Ac3IV, which is an interesting finding. However, some points need to be improved and clarified.

1. Little information on the glucagon and secondary antibodies used in this study. This is important information for multiple immunostainings. Please provide details.

2. In Fig.3D and Fig.7C/D, the degree of decrease in β-cell area by STZ administration and the degree of recovery by Ac3IV administration are quite different. Can this be explained by the difference in the mice used?

3. In Fig. 4C, the percentage of insulin+, GFP- cells increased to about 40% in Ac3IV administrated group, but in the representative image of Fig. 4A, the percentage of insulin+, GFP- cells does not seem to be 40%. Also, the sum of the cells in Fig.4C and D, STZ treated group exceeds 100%. What does the vertical axis represent? Moreover, insulin+ and GFP+ β-cells should account for a large percentage of the cells, this does not appear to be the case in the graphs of Fig. 4C-E. The authors should describe the analysis method and what the vertical axis represents in detail.

4. The authors do not discuss the mechanisms by which Ac3IV treatment increases alpha cell apoptosis and decreases proliferation. The authors should discuss it.

5. The values of the vertical axis in Fig.7G and H are much larger than those in Fig.7A and B. I wonder why the area of large islets (>25000 μm2) is larger than the islet area shown in Fig.7A and B. The authors should explain in detail how to analyze the graph and what the vertical axis represents.

6. Different results are shown between the head and tail of the pancreas regarding the effects of AC3IV, such as glucagon content and islet size distribution. Is there any relationship between the distribution of receptors or other factors that could be the reason for these results? Please discuss it.

6. PLOS authors have the option to publish the peer review history of their article (what does this mean?). If published, this will include your full peer review and any attached files.

Reviewer #1: No

Reviewer #2: No

---

## [Author Response · Author response to Decision Letter 0]

2 Sep 2021

Beneficial impact of Ac3IV, an AVP analogue acting specifically at V1a and V1b receptors, on diabetes islet morphology and transdifferentiation of alpha- and beta-cells by Mohan et al. PONE-D-21-20871

Reviewer 1:

We thank the Reviewer for assessment of our manuscript, in particular his/her comments that ‘It is new important knowledge that V1a and V1b receptor activation are related to improvement of both transdifferentiation of both alpha and beta cell’. Our responses to the queries raised by the Reviewer are considered below.

Specific comments

1. The Reviewer comments ‘The authors measured blood glucose and insulin levels as the effect of Ac3 IV on blood glucose level, glucose tolerance test and insulin tolerance test should be performed to evaluate glucose metabolism in more detail’. The authors thank the Reviewer for this comment. We recognise that glucose and insulin tolerance tests are useful for evaluating overall beneficial effects of Ac3IV on metabolism. However, the primary objective of the current study was to determine the impact of Ac3IV on islet morphology, and especially transdifferentiation of alpha- and beta-cells, in our two transgenic mouse models. Thus, we have already documented positive actions of Ac3IV on glucose disposal and insulin action in HFF mice in our previously published report (please see Diabetes Obes Metab. 2021 Jun 9. doi: 10.1111/dom.14462). Also, it is technically difficult to chart increases of circulating glucose after GTT in the present STZ model due to very high starting concentrations. To highlight these matters and address the Reviewers comment, we have made the following alterations to the text:

Introduction section (Page 4, lines 13-14): ‘In this way the impact of Ac3IV on alpha- to beta-, as well as beta- to alpha-cell, transdifferentaition could be investigated, which was the primary objective of the current study’.

Discussion section (Page 14, lines 16-20): ‘Assessment of glucose and insulin tolerance would also have been useful to determine the positive impact of Ac3IV on overall metabolism, but our primary objective in the transgenic mouse models utilised for the current study was evaluation of effects on islet cell lineage. Moreover, benefits of Ac3IV on glucose homeostasis and insulin action have previously been confirmed in HFF obese mice [5]’.

2. The Reviewer remarks ‘The authors used insulin and glucagon-positive cells to evaluate transdifferentiation between alpha and beta cells, but in order to obtain more detailed results, other markers such as PDX1, Nkx6.1, and Arx Expression should be assessed by staining or gene expression’. The authors thank the Reviewer for this comment. At present we are unfortunately unable to conduct the suggested experiments. Thus, for gene expression studies in the different alpha- and beta-cell populations, namely insulin+ve, GFP+ve, insulin+ve, GFP-ve, insulin-ve, GFP+ve , glucagon+ve, GFP-ve cells as well as glucagon+ve, GFP+ve cells, we would need to use cell sorting technologies that are beyond our current expertise. We are also somewhat reluctant to employ immunohistochemical staining approaches for the accurate determination of activation and levels of endogenous transcription factors. In addition, these are only predicted markers of cellular phenotype whereas we have employed a much more definitive cell lineage tracing approach using transgenic mice. However, to address the Reviewers point, we have added the following sentence to the Discussion section (Page 16, lines 12-15) that reads as follows: ‘Furthermore, although beyond the scope of the current study, consideration of levels of specific alpha- and beta-cell transcription factors such as aristaless-related homeobox (Arx), pancreatic and duodenal homeobox 1 (Pdx-1) or NK6 homeobox 1 (NKX6.1) would be useful to further assess islet cell lineage fate [7]’. 

Reviewer 2:

We thank the Reviewer for their appraisal of our manuscript, in particular his/her comments that ‘The authors suggest that Ac3IV administration improves the islet morphological changes induced by streptozotocin by Ac3IV, which is an interesting finding’. Our responses to the queries raised by the Reviewer are considered below.

Specific comment

1. The Reviewer comments ‘Little information on the glucagon and secondary antibodies used in this study. This is important information for multiple immunostainings. Please provide details.’. The authors thank the Reviewer for this comment, and we are happy to comply with his/her suggestion. We have added the following additional information relating to secondary antibodies employed to the Materials and Methods section of the revised manuscript (Page 6, lines 21-23 and Page 7, lines 2-7) that reads as follows: ‘The glucagon primary antibody (PCA2/4) was raised in house in guinea-pigs immunised with porcine glucagon-carbodiimide-albumin conjugates [21], and specificity confirmed in our previous studies [22] .……..... Slides were then rinsed in PBS and incubated for 45 min at 37°C with appropriate Alexa Fluor secondary antibodies (1:400; Invitrogen, Alexa Fluor 488 for green or 594 for red, Invitrogen). The following secondary antibodies were employed, as appropriate, goat anti-mouse Alexa Fluor 488, goat anti-mouse Alexa Fluor 594, goat anti-guinea pig Alexa Fluor 488, goat anti-guinea pig Alexa Fluor 594, donkey anti-goat 488 and goat anti-rabbit Alexa Fluor 488’. The reference section has been updated accordingly.

2. The Reviewer remarks ‘. In Fig.3D and Fig.7C/D, the degree of decrease in β-cell area by STZ administration and the degree of recovery by Ac3IV administration are quite different. Can this be explained by the difference in the mice used?’. The authors thank the Reviewer for this interesting comment and insightful suggestion. Thus, we have reported previously a more severe STZ-induced diabetic phenotype in Ins1CRE/+;Rosa26-eYFP when compared GluCreERT2;Rosa26-eYFP mice (Lafferty et al. Front Endocrinol (Lausanne). 2021; 12:633625). This was suggested to occur because of variations in STZ susceptibility between the strains of mice. The observation fully accords with greater reductions in islet and beta-cell areas observed in the current study in Ins1CRE/+;Rosa26-eYFP (Figure 3B,D) when compared to GluCreERT2;Rosa26-eYFP STZ-diabetic mice (Figure 7A-D). To follow up on the Reviewer’s comment, we have added the following text to the Discussion (Page 16, lines 8-12), that reads as follows: ‘However, the established detrimental effects of STZ on islet architecture were apparent to a greater extent in Ins1CRE/+;Rosa26-eYFP mice, in agreement with recent observations of a more severe STZ-induced phenotype in these mice compared with GluCreERT2;Rosa26-eYFP diabetic mice [15]. This difference was suggested to be linked to variations in STZ susceptibility between the two strains of mice’. 

3. The Reviewer comments ‘In Fig. 4C, the percentage of insulin+, GFP- cells increased to about 40% in Ac3IV administrated group, but in the representative image of Fig. 4A, the percentage of insulin+, GFP- cells does not seem to be 40%. Also, the sum of the cells in Fig.4C and D, STZ treated group exceeds 100%. What does the vertical axis represent? Moreover, insulin+ and GFP+ β-cells should account for a large percentage of the cells, this does not appear to be the case in the graphs of Fig. 4C-E. The authors should describe the analysis method and what the vertical axis represents in detail’. The authors thank the Reviewer for this comment and the opportunity to clarify these matters. We agree that the image displayed in Figure 4A is not a perfect representation of approximately 40% insulin+ve, GFP-ve cells in Ac3IV treated mice. We have now replaced this image with a more representative picture of the islets. With regards to the sum of the cells in Figure 4C&D, in panel C, the population of cells that express insulin only are being assessed, whereas as in panel D we are quantifying a different population of cells that only express GFP. As such, these are distinct populations of cells that we would not expect to add up to 100% in total for each treatment group. To help simplify this matter, we have now added small titles above each figure panel to make these graphs easier to interpret. Panel C is now labelled ‘Insulin-expressing non-beta cells’, panel D ‘Insulin-deficient beta-cells’ and panel E ‘Glucagon expressing beta-cells’. Finally, we fully agree that insulin+ve, GFP+ve cells should account for a large percentage of islet cells, and this is in fact the case. To note, we have not specifically reported on numbers of insulin+ve, GFP+ve cells in Figure 4, as this is simply the inverse of insulin-ve, GFP+ve (Figure 4D) cells. For example, in panel D the saline treatment group expresses around 20% insulin-ve, GFP+ve beta-cells, which implies that the remaining 80% of GFP+ve would be insulin+ve. We are confident that the new titles above each of the panels in Figure 4 will help with overall interpretation of these data. The authors thank the Reviewer for the chance to improve the quality and clarity of our figures.

4. The Reviewer notes ‘The authors do not discuss the mechanisms by which Ac3IV treatment increases alpha cell apoptosis and decreases proliferation. The authors should discuss it’. The authors thank the Reviewer for this comment and are happy to comply with his/her suggestion. In general, activation of V1a and V1b receptors is linked to cell pro-survival effects, both in normal islet cells [Biochimie. 2019; 158:191-198] as well as in epithelial cells, glomerular mesangial cells and vascular smooth muscle [Front Med (Lausanne). 2015; 24:19]. This is suggested to be linked to V1a and V1b Gq receptor coupling and activation of the phospholipase C (PLC) pathway. In disease states such as diabetes, combined V1a and V1b receptor activation has been shown to result in decreased proliferation of both pancreatic alpha- and beta-cells [Diabetes Obes Metab. 2021 Jun 9. doi: 10.1111/dom.14462]. In keeping with this, the impact of Ac3IV to increase apoptosis and decrease proliferation of alpha-cells, most likely relates to the disease phenotype induced by STZ administration in the current setting. As such, STZ administration is well established to reduce islet beta-cell mass and dramatically increase alpha-cell mass. Therefore, the impact of Ac3IV on alpha-cells most likely reflects promotion of a more normal islet architecture. To better highlight the matter, we have added the following sentence to the Discussion section of the revised text, that reads (Page 13, lines 14-18): ‘More notably, the decreases of alpha-cell area and pancreatic glucagon were accompanied by enhanced alpha-cell apoptosis and tendency for decreased alpha-cell proliferation. This is interesting given that V1a and V1b receptor activation is generally associated with pro-survival effects [2], linked to Gq receptor coupling and activation of the phospholipase C (PLC) pathway. Effects of Ac3IV in the current setting may simply represent the capacity for Ac3IV to reverse the detrimental effects of STZ on pancreatic islet architecture. Indeed, similar effects of Ac3IV to decrease alpha-cell growth have been demonstrated in HFF mice [5]’.

5. The Reviewer comments ‘The values of the vertical axis in Fig.7G and H are much larger than those in Fig.7A and B. I wonder why the area of large islets (>25000 μm2) is larger than the islet area shown in Fig.7A and B. The authors should explain in detail how to analyze the graph and what the vertical axis represents’. We thank the Reviewer for the opportunity to explain this point. He/she is correct that our original graph axes labelling, and subsequent presentation of data, could lead to some confusion within Figure 7. We apologise for this, but are grateful that we can now correct the matter. In the original version of Figure 7, panels G&H, islet size distribution data was simply displaying the average size of small (< 10,000 µm2), medium (10,000 – 25,000 µm2) and larger (> 25,000 µm2) sized islets, with no indication of the percentage of total islets within each islet size group. We have now converted these data to depict a percentage of total islets analysed. We are confident that this change has addressed the Reviewers concern. 

6. The Reviewer further comments ‘Different results are shown between the head and tail of the pancreas regarding the effects of AC3IV, such as glucagon content and islet size distribution. Is there any relationship between the distribution of receptors or other factors that could be the reason for these results? Please discuss it’. We thank the Reviewer for this comment and the opportunity to further discuss this matter. As noted in the original manuscript, pancreatic islets are more concentrated and richer in alpha-cells in the tail, as opposed to the head, of the pancreas. To the best of our knowledge, there is no definitive literature to suggest differences in receptors or other factors between the head and tail portions of the pancreas. However, in the context of the current study and islet cell lineage tracing, it may be of interest to note that the head and the tail of the pancreas have different developmental origins. Thus, the head of the pancreas is formed from the dorsal and ventral pancreatic bud, and the tail from the ventral pancreatic bud only. In addition, previous studies in rodents have shown that islets originating from the dorsal pancreatic bud have greater capacity to secrete and synthesise insulin than islets of ventral bud origin. We have therefore added the following sentences to the Discussion section of the revised manuscript in relation to this information, that read as follows (Page 15, lines 7-12): ‘Furthermore, in terms of effects of Ac3IV on islet cell lineage it may be of interest to note that the head and the tail of the pancreas have different developmental origins. As such, the head of the pancreas is formed from the dorsal and ventral pancreatic bud, whereas the tail is derived from the ventral pancreatic bud [30]. Moreover, there is a suggestion that islets from the head of the pancreas contain more glucagon and somatostatin and have a greater insulin secretory capacity than those originating solely from the ventral bud region [31]’. The reference section has been updated accordingly.

---

## [Decision Letter · Decision Letter 1]

22 Sep 2021

PONE-D-21-20871R1Beneficial impact of Ac3IV, an AVP analogue acting specifically at V1a and V1b receptors, on diabetes islet morphology and transdifferentiation of alpha- and beta-cellsPLOS ONE

Dear Dr. Irwin,

Thank you for submitting your manuscript to PLOS ONE. After careful consideration, we feel that it has merit but does not fully meet PLOS ONE’s publication criteria as it currently stands. Therefore, we invite you to submit a revised version of the manuscript that addresses the points raised during the review process.

Your manuscript has been evaluated by two reviewers. After careful consideration, we feel that it has merit but does not fully meet PLOS ONE’s publication criteria as it currently stands, because experiments are described in sufficient detail. Authors should address concerns suggested by the reviewer for further consideration in PLoS ONE. 

We look forward to receiving your revised manuscript.

Kind regards,

Wataru Nishimura, M.D., Ph.D.

Academic Editor

PLOS ONE

Journal Requirements:

Reviewers' comments:

Reviewer's Responses to Questions

**Comments to the Author**

1. If the authors have adequately addressed your comments raised in a previous round of review and you feel that this manuscript is now acceptable for publication, you may indicate that here to bypass the “Comments to the Author” section, enter your conflict of interest statement in the “Confidential to Editor” section, and submit your "Accept" recommendation.

Reviewer #1: All comments have been addressed

Reviewer #2: (No Response)

2. Is the manuscript technically sound, and do the data support the conclusions?

Reviewer #1: Yes

Reviewer #2: Yes

3. Has the statistical analysis been performed appropriately and rigorously? 

Reviewer #1: Yes

Reviewer #2: Yes

4. Have the authors made all data underlying the findings in their manuscript fully available?

Reviewer #1: Yes

Reviewer #2: (No Response)

5. Is the manuscript presented in an intelligible fashion and written in standard English?

Reviewer #1: Yes

Reviewer #2: (No Response)

6. Review Comments to the Author

Reviewer #1: (No Response)

Reviewer #2: Thank you to the authors for their careful consideration and answers to the previous comments. The manuscript has greatly improved, and I only have one but an important comment.

Previous Comment No.5: Fig. 7G was made clearer by the authors' revision.

However, it seems that the intention of my question was not well understood by the authors.

Regarding Fig.7 G and H, the authors classified the islet into small (< 10,000 μm2), medium (10,000-25,000 μm2) and large (> 25,000 μm2) islets. However, in Fig.7A/B and Fig.3B, the graphs show that the area of islets and the maximum value is about 10,000 μm2. It is questionable that these areas are much smaller than the average area of a single small size islet.What does the islet area in Fig.7A/B and Fig.3B represent? If it represents the area of islets per unit area of pancreas, please describe it that readers can understand it.

7. PLOS authors have the option to publish the peer review history of their article (what does this mean?). If published, this will include your full peer review and any attached files.

Reviewer #1: No

Reviewer #2: No

---

## [Author Response · Author response to Decision Letter 1]

28 Sep 2021

Beneficial impact of Ac3IV, an AVP analogue acting specifically at V1a and V1b receptors, on diabetes islet morphology and transdifferentiation of alpha- and beta-cells by Mohan et al. PONE-D-21-20871R1

Reviewer 1:

We are pleased to note Reviewer 1 has no further comments and is happy with our revised manuscript.

Reviewer 2:

We thank the Reviewer for their reappraisal of our revised manuscript, and particularly the following comment: ‘Thank you to the authors for their careful consideration and answers to the previous comments. The manuscript has greatly improved’. Our response to the one remaining query raised by the Reviewer is considered below.

Specific comment

1. The Reviewer notes ‘I only have one but an important comment. Previous Comment No.5: Fig. 7G was made clearer by the authors' revision. However, it seems that the intention of my question was not well understood by the authors. Regarding Fig.7 G and H, the authors classified the islet into small (< 10,000 μm2), medium (10,000-25,000 μm2) and large (> 25,000 μm2) islets. However, in Fig.7A/B and Fig.3B, the graphs show that the area of islets and the maximum value is about 10,000 μm2. It is questionable that these areas are much smaller than the average area of a single small size islet. What does the islet area in Fig.7A/B and Fig.3B represent? If it represents the area of islets per unit area of pancreas, please describe it that readers can understand it’. The authors agree that on first look there may appear to be a small discrepancy between the islet areas depicted within panels A&B of Figure 7, versus panels G&H. We are pleased for the opportunity to clarify this matter and remove any ambiguity. By way of explanation, within the group of islets classified as small in Figure 7G&H, that is < 10,000 μm2 in area, a large proportion of these islets are actually less than 2,000 μm2. This accounts for the seemingly low overall average islet and beta cell areas depicted in Figure 7A&B. To make this more apparent, we have added an extra classification of ‘very small islets’, of < 2,000 μm2 in area, to our data analyses within Figure 7G&H. We are confident that this change has now addressed the Reviewers concern. To fully highlight this change we have made the following alterations to the Results section of the manuscript (Page 11, line 23 and Page 12, lines 1-6), that reads: ‘The number of very small (<2,000 µm2), small (2,000 - 10,000 µm2) and medium (10,000 - 25,000 µm2) sized islets was similar across all groups of mice in the head of the pancreas, but larger (>25,000 µm2) sized islets were only detectable in Ac3IV treated mice (Fig. 7G). Islet size distribution was not noticeably different between all the various groups in pancreatic tail, but Ac3IV treated mice did appear to have reduced numbers of very small (<2,000 µm2) islets, with larger sized islets being more prevalent (Fig. 7H)’. No further alterations to the text are required as overall interpretation of these data has not changed.

---

## [Editor Report · Decision Letter 2]

7 Dec 2021

Beneficial impact of Ac3IV, an AVP analogue acting specifically at V1a and V1b receptors, on diabetes islet morphology and transdifferentiation of alpha- and beta-cells

PONE-D-21-20871R2

Dear Dr. Irwin,

We’re pleased to inform you that your manuscript has been judged scientifically suitable for publication and will be formally accepted for publication once it meets all outstanding technical requirements.

Kind regards,

Wataru Nishimura, M.D., Ph.D.

Academic Editor

PLOS ONE
---

## [Editor Report · Acceptance letter]

10 Dec 2021

PONE-D-21-20871R2 

Beneficial impact of Ac3IV, an AVP analogue acting specifically at V1a and V1b receptors, on diabetes islet morphology and transdifferentiation of alpha- and beta-cells 

Dear Dr. Irwin:

I'm pleased to inform you that your manuscript has been deemed suitable for publication in PLOS ONE. Congratulations! Your manuscript is now with our production department. 

Kind regards, 

on behalf of

Dr. Wataru Nishimura 

Academic Editor

PLOS ONE